# Prediction Method of Blast Wave Impact on Crew Module for Liquid Rocket Explosion on Launch Pad

**Yan Wang \*, Hua Wang, Cunyan Cui and Beilei Zhao**

Space Engineering University, Beijing 101416, China; wanghuaprofessor@163.com (H.W.);
ccy6655@126.com (C.C.); zhaobeileistudent@163.com (B.Z.)

\* Correspondence: 08wy@163.com; Tel.: +86-010-132-6196-3676

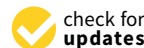

**Featured Application: The work can be used to estimate the impact on crew modules of blast wave caused by liquid rocket explosion, and to improve the ability of crew modules and to develop safety standards by using numerical simulations.**

**Abstract:** The role of manned space flight in the field of space exploration and utilization is growing. However, the security system of the manned spaceflight is still imperfect. In the case that the rocket explodes, crew modules maybe damaged by the blast wave, which will threaten the safety of the crews. This research aims to obtain the necessary data and information to enable the designers of the launch vehicles and crew modules to develop safer launch systems. To this end, this paper proposes a numerical method using LS-DYNA to study the propagation law of blast waves caused by rocket explosion on the launch pad and to quantify the impact of the blast wave on crew module. The numerical results indicate that the final blast waveform of the model with rocket is conical in the upper and lower parts, and spherical in the middle. At the same time, the third-stage explosion is the most harmful to the crew module, while the first-stage explosion is the least. Furthermore, the model with rocket has a marked effect on explosion strength: the pressure enhancement factor is about 4–17 times. Most importantly, overpressure prediction formula acting on the crew modulesof explosion on the launch pad is established for quick peak overpressure predicting and damage evaluating.

**Keywords:** manned spaceflight; crew module; blast wave; peak overpressure; launch pad; LS-DYNA; liquid propellant; pressure enhancement factor

---

## 1. Introduction

Manned spaceflight is playing an important role in the exploration and utilization of space. However, research on the safety systems in the whole process of space exploration is relatively scarce. The launch vehicle is a complex and large-scale technology-intensive system. It plays an important role in the manned space mission. Once an explosion occurs, it would threaten the safety of crew in the crew modules of rockets. Generally speaking, rocket explosion can be divided into various modes such as explosion on launch pad, in flight and impacting to ground, and this paper focuses on the explosion mode on the launch pad.

The study of rocket blast environment is an ongoing effort to characterize the environment resulting from catastrophic rocket explosion. The purpose is to develop the data and information required to allow launch vehicle and crew system designers to develop safer crewed launch systems. A massive explosion of a liquid-propellant rocket in the course of an accident can lead to a truly catastrophic event. Owing to its very low boiling point, liquid propellant is extremely hazardous and its leakage/sudden release from a pressurized tank would lead to vapor cloud formation resulting in subsequent fire and explosion.

This point has been amply demonstrated by a series of explosion accidents over the past decades. The space shuttle Challenger exploded 73 s after launch in 1986, killing all seven astronauts on board. As was established by the Challenger investigation, the original source of the disaster was freezing of the O-ring in the lower section of the left solid booster and formation of a gas leak through the O-ring [1]. The Challenger disaster provoked studies of various risks that can lead to similar catastrophic events. In September 2016, the approximately 549 t Falcon-9 rocket exploded on the launch pad, causing $200 million worth of damage to the AMS-6 communications satellite. Moreover, Space X's main launch pad at the Cape Canaveral satellite launch base, severely affecting subsequent launch plans. The most likely cause of the accident was a leak of the helium system in the oxygen tank of upper stage. In April 2015, a Falcon-9 carrying 2 t cargo exploded just 139 s after launch, causing rocket destruction and delay in subsequent launch plans.

The hazards to personnel and facilities around the launch site must be considered in safety analysis and assessment. Many workers studied liquid propellant explosion characteristics by conducting experiments and analyzing experimental data. Between 1963 and 1969, NASA and other units conducted a series of rocket explosion scaled tests using liquid propellants such as liquid oxygen/kerosene, liquid hydrogen/liquid oxygen, and studied the hazardous characteristics of liquid propellant explosion [2–4]. From 1962 to 1969, ADL (Advanced Distributed Learning) Corporation and the MIT (Massachusetts Institute of Technology) et al. managed the liquid oxygen/kerosene propellant tests of Saturn rockets during simultaneous mixing and separate mixing, to study the explosive yields of liquid propellant [5]. Between 1966 and 1968, Fletcher et al. reviewed the hazards of liquid propellant explosion blast wave and determined the basic methods for blast wave research and hazard assessment [6]. To develop a better understanding of the magnitude and the character of explosions resulting from a breach of LH2/LOX tanks and to clarify the important questions about cryogenic H2/Ox explosion mechanisms, Hydrogen-Oxygen vertical-impact (HOVI) tests were carried out at the NASA Johnson Space Center's White Sands test facility [7]. NASA had been funding an ongoing development program to characterize the explosive environment produced by using the test and accident data available from public or NASA sources as well as focused scaled tests that are focusing on the near field environment that threaten the crew modules [8]. An empirical non-TNT approach was established to predict launch vehicles explosion environment characteristics in the near field by James. All available empirical data from full scale launch vehicle explosion tests and accidents was assembled. Crew survivability from catastrophic explosions could be increased in this way [9].

Compared with test research, numerical simulation methods have some advantages, including reducing costs, improving efficiency, and facilitating parameters analysis. Osipov et al. conducted the assessment based on an interpretation of these data via analytical semiquantitative estimates and numerical simulations of simplified models for the whole range of the physical phenomenon governing the outcome of a propellant-tank breach [10]. Hosangadi et al. assessed a specific failure resulting from a catastrophic disintegration during ascent of the crew launch vehicle liquid propellant tank and determined the time scales of the various processes and the strength and propagation of the blast wave if the mixture ignites [11]. Jon et al. presents air blast and fragmentation hazard analyses during processing of launch vehicle in the Vehicle Assembly Building as well as consequence and risk analyses to provide added information for the decision maker [12]. Lambertet al. studied explosive yields for Delta heavy vehicles in the explosion mode of impacting the ground with different distances and angles [13].

Explosion is a complex multidisciplinary problem that involves combustion, fluid dynamics, thermodynamics, structure/fluid interaction, and shock physics for high-speed flows, especially the accidents. The physical process governing the breakup and vaporization of the bulk liquid propellants are extremely complex with a variety of phenomena such as flashing, aerodynamic gas/liquid mixing, and vaporization occurring concurrently. The launch vehicle makes this even more complex in that the vehicle is accelerating upward with the atmospheric conditions changing with altitude and vehicle velocities. The TNT (2,4,6-Trinitrotoluene) equivalent model is a good method using worst-case

assumptions to define the potential explosions risk in an attempt to envelope the blast environment for crew module

The liquid propellant explosion test has high risk and high cost, and the experimental conditions are strict and the repeatability is poor, which makes the test difficult. With the development of computer simulation technology, numerical simulation has become the main method to solve such problems [14].

Blast loads in simple geometries can be predicted by using empirical formulas. In a more complex model with rocket, one approach is to use numerical tools such as LS-DYNA [15] to provide reliable estimates of the effects of adjacent structures on the blast loads. Numerical simulations can be used to extend the database of blast effects on the crew modules. Experiments have been conducted to validate the numerical simulation techniques against blast load experimental results. Numerical simulations can help to understand the hazards to the crew module of the blast wave. While detailed experimental investigations are not easy to perform, numerical simulations can close that gap. Hence, the number of field experiments can also be reduced by using numerical simulations. While an experimental investigation captures mainly one scenario, numerical simulations can be used to investigate a multiplicity of threat scenarios [16].

Bo et al. simulated the peak overpressure and attenuation law of blast wave of liquid propellant explosion in the air by using AUTODYN software [17]. Ke and Jingpeng et al. used the accident consequence analysis software PHAST to study rocket explosion hazards on launch pad, predicting the scope of the rocket explosion hazards, and dividing the safety distance with the relevant criteria [18,19]. However, hazards for the rocket explosion on launch pad and the crew module exposed to the blast wave have received relatively little attention. The results achieved previously cannot be used to estimate the impact on crew modules of blast wave caused by liquid rocket explosion on launch pad because potential hazards, including explosion overpressures, have not been clear. Therefore, it is important and far-reaching to study the propagation law of blast wave caused by rocket explosion on the launch pad and quantify the impact on the crew modules of blast wave to improve the ability of crew modules and to develop safety standards by using numerical simulations.

## 2. Explosion Mode Analysis and Typical Scenario Setting

By collecting the typical explosion accidents all over the world in recent decades, the explosion modes are classified. This paper focuses on the study of the explosion mode on the launch pad, and typical accident statistics are shown in Table 1. Based on the analysis of typical accident cases of rocket explosion mode on launch pad, four typical scenarios are put forward. Through the numerical simulation analysis of four typical explosion scenarios, the rocket explosion characteristics under this mode are studied. The process descriptions and propellants involved in the explosion of the four typical scenarios are shown in Table 2.

**Table 1.** Typical accident statistics of rocket explosion on launch pad.

| Time | Country | Rocket | Mass/t | Why &How | Location | Consequences |
|------|---------|--------|--------|----------|----------|--------------|
| 1960 | Soviet Union | R16 | 141.2 | In the preparation stage, the engine did not ignite as ordered. When technicians approached the rocket for fault inspection, the engine suddenly ignited and exploded. | stage-2 | Rocket and payload are destroyed, 165 deaths |
| 1980 | Soviet Union | Vostok | 279 | During the propellant refilling and final testing, the rocket exploded due to a fuel leak. | unknown | Rocket and payload are destroyed, 50 deaths |

**Table 1.** *Cont.*

| Time | Country | Rocket | Mass/t | Why &How | Location | Consequences |
|---|---|---|---|---|---|---|
| 1983 | Soviet Union | Soyuz | 303 | One of the boosters exploded before the launch, causing a full rocket to explode. | booster | Rocket and payload are destroyed, astronauts escape successfully |
| 2003 | Brazil | VLS | 50 | During the pre-launch maintenance test, an engine misfire caused an explosion. | stage-1 | Rocket and payload are destroyed, launch pad destroyed, 21 deaths, more than 20 people were seriously injured |
| 2016 | USA | Falcon9 | 549 | The cryogenic helium system in the secondary oxygen tank leaked and caused an explosion during the pre-launch test and refuelling. | stage-2 | Rocket and payload are destroyed, launch pad destroyed, no casualties |

**Table 2.** Scenarios setting of the rocket explosion on launch pad.

| Scenarios | Process Description |
|---|---|
| 1 | Structure destruction of propellant tank of stage-3 causes propellant leakage and explosion, which results in the structure destruction of the propellant tank of the stage-1, stage-2 and boosters and the explosion. |
| 2 | Structure destruction of stage-2 propellant tank causes propellant leakage and explosion, which results in the structure destruction of the propellant tank of the stage-1 and boosters and simultaneous explosion of the vertically falling stage-3. |
| 3 | Structure destruction of propellant tank of stage-1 causes structure destruction of booster tank, which results in stage-2 and stage-3 falling vertically, and then all propellants explode at the same time |
| 4 | Structure destruction of propellant tank of stage-1 causes propellant leakage and explosion, which leads to stage-2 falling vertically and explosion with boosters, and then stage-3 falls vertically and explodes. |

## 3. Definite of Explosive Yield

### 3.1. Definite Method of Explosive Yield forLiquid Propellant

TNT equivalent model is to convert the explosive materials into equivalent TNT according to the equal energy, and then the explosive law of TNT is applied to predict the effect of liquid propellant explosion [20]. It is widely used in the field of explosive characteristics of liquid propellant all over the world because of the ease of use [21]. Given that the mass of liquid propellant $M_0$ and explosive yield $Y$ of different modes, the TNT $M_T$ can be calculated by using Equation (1).

$$M_T = Y{\cdot}M_0 \tag{1}$$

where, $M_T$ represents mass equivalent to TNT, kg. $Y$ stands for explosive yield, dimensionless. $M_0$ is the total mass of the liquid propellant, kg.

It is very difficult to obtain accurate explosive yield theoretically under different explosion modes, because the chemical reaction mechanism of liquid propellant is different from that of solid explosive, and is affected by various factors such as the type, mass, detonation time of propellant,

falling speed and ground property, etc. In order to study the explosive yield in different explosion mode, explosive tests were carried out on different propellants of different masses in the United States [2–4]. Through statistical analysis of test data, explosive yield of liquid propellant is estimated. Explosive yield estimation is mainly calculated by two methods, that is, table-lookup-method and chart-reading-method. Therefore, explosive yield under the two methods is calculated respectively and it is finally determined according to the minimum principle [22].

Table-lookup-method is to obtain the explosive yield by reading Table 3, on the basis of determining the propellant type and the explosion mode. While as for thechart-reading-method, explosive power can be obtained according to propellant mass, and the final explosive yield is obtained by multiplying the specific coefficients corresponding to different propellants. The explosive yields of propellant for each stage of a rocket calculated by the two methods are shown in Table 4.

**Table 3.** Explosive yields and specific coefficients of different liquid propellant combination under different explosion modes.

| Propellant Combination | Explosion on Launch Pad | Explosion Outside Launch Pad | Mix Proportion | Specific Coefficient |
|---|---|---|---|---|
| $LOX/LH_2$ | 60% | 60% | 1:5 | 370% |
| LOX/RP-1 | Within 226.7995t is 20% plus 10% for rest part | 10% | 1:1.25 | 125% |
| $N_2O_4/UDMH$ | 10% | 5% | 1:2 | 240% |

**Table 4.** Equivalent yields at different stages of a certain rocket.

| Items | | Boosters | | Stage-1 | | Stage-2 | | Stage-3 | |
|---|---|---|---|---|---|---|---|---|---|
| Propellant | | $N_2O_4$ | UDMH | $N_2O_4$ | UDMH | $N_2O_4$ | UDMH | $LO_2$ | $LH_2$ |
| Propellant mass (kg) | | $4 \times 2.77 \times 10^4$ | $4 \times 1.35 \times 10^4$ | $1.26 \times 10^6$ | $6.07 \times 10^4$ | $3.38 \times 10^4$ | $1.60 \times 10^4$ | $1.53 \times 10^4$ | $3.16 \times 10^3$ |
| Total mass (kg) | | $1.65 \times 10^5$ | | $1.87 \times 10^5$ | | $4.98 \times 10^4$ | | $1.84 \times 10^4$ | |
| | Table-lookup-method | 0.10 | | 0.10 | | 0.10 | | 0.60 | |
| Y | Chart-reading-method | 0.19 | | 0.19 | | 0.20 | | 0.74 | |
| | Chosen | 0.10 | | 0.10 | | 0.10 | | 0.60 | |

## 3.2. ExplosiveYieldforDifferent Scenarios

The object of this paper is a three-stage rocket with four boosters, whose main size parameters of a certain rocket are shown in Figure 1. Combined explosive yields and combined explosive center coordinates are acquired by Equations (2) and (3), on the basis of the explosion yields of each stage. According to the definite method of explosive yield above, the explosion yield, combined explosion yield and the distance $R$ between the explosive center and crew modules of different scenes are listed in Table 5. Scaled distance is defined as $\overline{R} = R/M_T^{1/3}$.

$$y_{zh} = \sum m_i y_i / \sum m_i \qquad (2)$$

where $y_{zh}$ is the combined explosive yield, $y_i$ is the explosive yield of stage $i$, $m_i$ is total propellant mass of stage $i$, whose unit is kg.

$$x_z = \sum m_i x_i / \sum m_i \qquad (3)$$

where, $x_i$ is the distance between the middle of tanks of stage $i$ and the explosive center, whose unit is m.

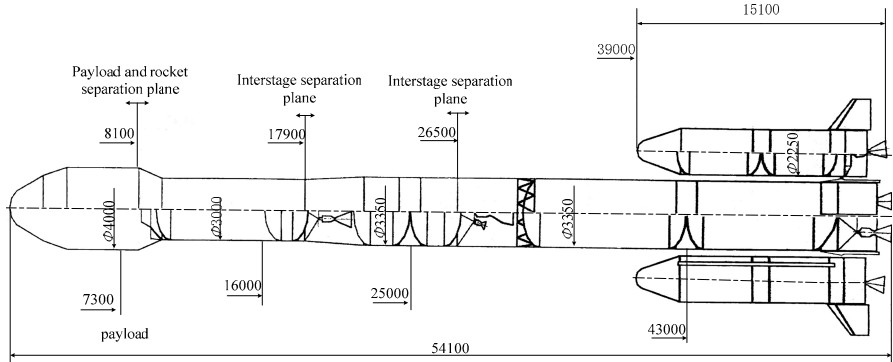

**Figure 1.** Main size parameters of a certain rocket.

**Table 5.** Parameters for typical scenarios of rocket explosion on launch pad.

| Scenarios | StagesInvolved in Explosion | Propellant Mass(kg) | Combined Explosive Yield | Equivalent TNT(kg) |
|---|---|---|---|---|
| 1 | stage-3 | $1.84 \times 10^4$ | 0.60 | $1.11 \times 10^4$ |
| | stage-1, 2 and boosters | $4.01 \times 10^5$ | 0.10 | $4.01 \times 10^4$ |
| 2 | stage-2 | $4.98 \times 10^4$ | 0.10 | $4.98 \times 10^3$ |
| | stage-1, 3 and boosters | $3.70 \times 10^5$ | 0.12 | $4.62 \times 10^4$ |
| 3 | whole rocket | $4.19 \times 10^5$ | 0.12 | $5.12 \times 10^4$ |
| | stage-1 | $1.87 \times 10^5$ | 0.10 | $1.87 \times 10^4$ |
| 4 | stage-2 and boosters | $2.14 \times 10^5$ | 0.10 | $2.14 \times 10^4$ |
| | stage-3 | $1.84 \times 10^4$ | 0.60 | $1.11 \times 10^4$ |

## 4. Finite ElementModel of Rocket Explosion

### 4.1. Computational Models and Algorithms

The impact on the crew module of blast wave under rocket explosion on launch pad could be studied through LS-DYNA. ANSYS LS-DYNA is an explicit analysis tool for modeling nonlinear dynamics of solids, fluids, gas, and their interaction. The 1:1 finite element analysis model including rocket body (including three stages and boosters), crew module, TNT explosive and air domain is set up by LS-DYNA software. In order to improve accuracy, air domain which is a ø20 m × 57.4 m cylinder and crew module which is a ø3 m × 0.1 m cylinder are established. Considering the symmetry of rocket, for the sake of saving computation time, a 1/4 symmetrical geometrical model of rocket is adopted.The liquid propellant in the rocket is equivalent to a cylindrical TNT, whose size varies with the different explosion scenes of the rocket, as shown in Table 6. Rocket 3D finite element model and gauges distribution are shown in Figure 2 the thickness of rocket skin is 0.003 m and other size of rocket can be read in Figure 2 [23].

It is assumed that the radius of TNT is $r_0$ (m) and the height is $h_0$ (m). Therefore, the length-diameter ratio of TNT is $\varphi = h_0/2r_0$.Given the mass of TNT, $\varphi$ of TNT can be calculated by using Equation (5) which is derived from the three equations in the Equation (4). Size parameter settings for different explosion scenarios are showed in Table 6. Unit system of cm-g-ms is adopted in the numerical model. The explosive source is located at the center of mass and a point-detonation-method is used [24].

$$\begin{cases} M_T = \rho \cdot V \\ V = \pi r_0^2 h_0 \\ \varphi = h_0/2r_0 = 1 \end{cases} \tag{4}$$

$$r = \sqrt[3]{\frac{M_T}{4\pi\rho}} \tag{5}$$

**Table 6.** Parameters setting of TNT model in numerical simulation.

| Scenarios | Stages Involved in Explosion | Equivalent(kg) | Radius(m) | Height (m) | Detonation Coordinate ($x_0$, $y_0$, $z_0$)/m |
|---|---|---|---|---|---|
| 1 | stage-3 | $1.11 \times 10^4$ | 0.81 | 3.26 | (0, 0, 32.57) |
| | stage-1, 2 and boosters | $4.01 \times 10^4$ | 1.25 | 5.00 | (0, 0, 10.07) |
| 2 | stage-2 | $4.98 \times 10^3$ | 0.62 | 2.49 | (0, 0, 27.57) |
| | stage-1, 3 and boosters | $4.62 \times 10^4$ | 1.31 | 5.24 | (0, 0, 27.57) |
| 3 | whole rocket | $5.12 \times 10^4$ | 1.36 | 5.43 | (0, 0, 47.57) |
| 4 | stage-1 | $1.87 \times 10^4$ | 0.97 | 3.88 | (0, 0, 11.57) |
| | stage-2 and boosters | $2.14 \times 10^4$ | 1.02 | 4.06 | (0, 0, 10.57) |
| | stage-3 | $1.11 \times 10^4$ | 0.81 | 3.26 | (0, 0, 32.57) |

In the finite element model (Figure 2), the eight-node element of SOLID 164 is adopted for the 3D explicit analysis. In order to prevent the element distortion in large deformation and nonlinear structural analyses, an arbitrary Lagrangian-Eularian (ALE) algorism is used in this paper.TNT and air are modeled with ALE multi-material grids, but the rocket and crew module with Lagrangian grids, while the minimal time step is controlled by the smallest element size in the explicit integral method. The grid is divided by grid gradient method, with the minimum size of 0.05 m and maximum size of 0.30 m. Fluid-solid coupling algorithm is used between two grids [25].

Furthermore, the transitional displacement of the nodes normal to the symmetry planes is constrained, in order to simulate the propagation effect of blast wave in the symmetric plane. Non-reflecting boundary condition is applied to top, bottom, and lateral surfaces, which allows the fluid medium to flow out in order to simulate the effect of infinite space [26].

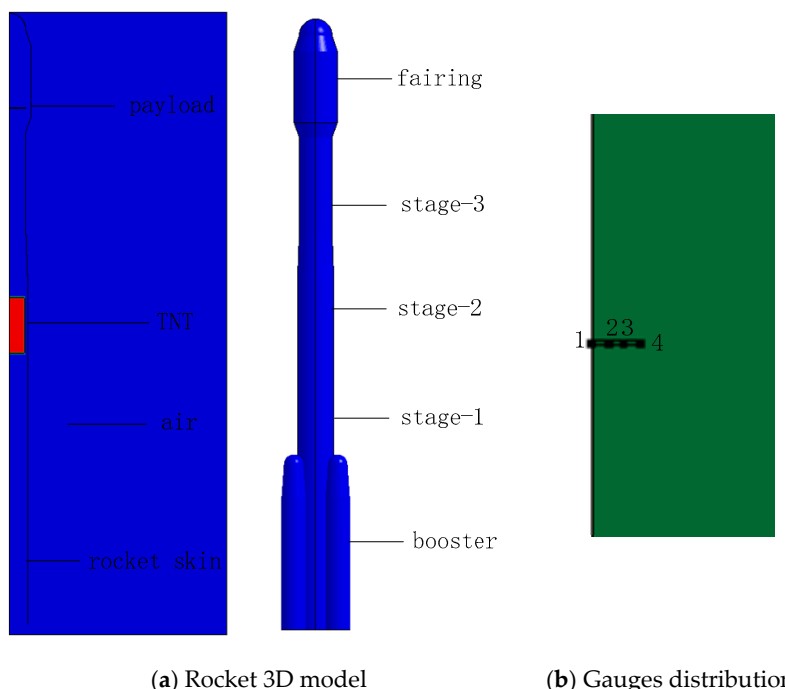

(**a**) Rocket 3D model　　　　　　　(**b**) Gauges distribution

**Figure 2.** Rocket 3D finite element model and gauges distribution.

### 4.2. Material Constitutive Models and State Equations

Three kinds of materials are involved in this finite element model: air, TNT, aluminum alloy.

The air is commonly modeled by null material model with a linear polynomial equation of state, EOS_LNIEAR_POLYNOMIAL, which defines the pressure by the following equation:

$$P = C_0 + C_1\mu + C_2\mu^2 + C_3\mu^3 + \left(C_4 + C_5\mu + C_6\mu^2\right)E_0 \tag{6}$$

where the parameter $\mu$ is defined as $\mu = \rho/\rho_0$, $\rho$ is the current density, and $\rho_0$ is a nominal or reference density; $C_0$–$C_6$ are the equation coefficients; and the parameter $E_0$ is the initial internal energy of reference specific volume per unit.

Table 7 gives the parameters used in the air model by Liu, M.Y. and Hu, Q. Refs. [27,28].

**Table 7.** Parameters of air.

| $\rho$ (g/cm$^3$) | $C_0$ | $C_1$ | $C_2$ | $C_3$ | $C_4$ | $C_5$ | $C_6$ | $E_0$ (J/m$^3$) | $V_0$ |
|---|---|---|---|---|---|---|---|---|---|
| 1.29 | $-1.0 \times 10^{-6}$ | 0 | 0 | 0 | 0.4 | 0.4 | 0 | 25 | 1.0 |

The TNT charge is modeled by the high explosive material model, MAT-HIGH-EXPLOSLVE-BURN, and the Jones-Wilkins-Lee (JWL) equation of state defines the pressure by the following equation:

$$P = A\left(1 - \frac{\omega}{R_1 V}\right)e^{-R_1 V} + B\left(1 - \frac{\omega}{R_2 V}\right)e^{-R_2 V} + \frac{\omega E}{V} \tag{7}$$

where $A$, $B$, $R_1$, $R_2$, $\omega$ are the equation coefficients and they all should be tested in an accurate blast analysis; and $V$ is the initial relative volume.

Here we adopt the parameters selected by Shang X. [29], as shown in Table 8.

**Table 8.** Parameters of TNT charge.

| $\rho$ (g/cm$^3$) | $v_D$ (km/s) | $P_{CJ}$ (GPa) | $A$ (GPa) | $B$ (GPa) | $R_1$ | $R_2$ | $\omega$ | $E_0$ (J/m$^3$) | $V_0$ |
|---|---|---|---|---|---|---|---|---|---|
| 1.63 | 6.93 | 27 | 371 | 7.43 | 4.15 | 0.95 | 0.3 | $7 \times 10^3$ | 1.0 |

Various mathematical models have been adopted to represent the flow stress data over a range of temperatures and strain rates. Among all, the Johnson-Cook constitutive model is chosen for aluminum alloy [30] in the research, as it provides a good description of the metal material behavior, subjected to large strain, strain rates and high temperatures. The JC equation is given by

$$\sigma = \left(A + B\varepsilon_p^n\right)\left(1 + C\ln\dot{\varepsilon}^*\right)\left[1 - \left(\frac{T - T_{room}}{T_m - T_{room}}\right)^m\right] \tag{8}$$

where $\sigma$ is the flow stress, $\varepsilon$ is the equivalent plastic strain, $A$ is the yield stress of the material at the condition of reference temperature and reference strain, $B$ is the strain hardening coefficient, $n$ is the strain hardening exponent, $C$ and $m$ are the material constants representing the coefficient of strain rate hardening and thermal softening exponent, respectively. $\dot{\varepsilon}^*$ is the dimensionless strain rate and can be expressed as $\dot{\varepsilon}^* = \dot{\varepsilon}/\dot{\varepsilon}_r$, where $\dot{\varepsilon}$ is the strain rate and $\dot{\varepsilon}_r$ is the reference strain rate. $T$ is the deformation temperature, $T_m$ is the melting temperature of the investigated material and $T_{room}$ is the reference temperature, all of which should be absolute temperature (K) [31]. The three items on the right side of Equation (8) are used to describe the phenomena of the work-hardening effect, strain-rate effect and temperature effect, respectively. The equation of state for aluminum alloy is described by Gruneisen [32]. Detailed parameters are set out in Table 9.

**Table 9.** Parameters of aluminum alloy.

| $\rho$ (g/cm$^3$) | $G$ (GPa) | $A$ | $B$ | $n$ | $C$ | $m$ | $T_m$ (K) | $T_{room}$ (K) |
|---|---|---|---|---|---|---|---|---|
| 2.75 | 47 | $2.65 \times 10^{-3}$ | $4.26 \times 10^{-3}$ | $3.4 \times 10^{-1}$ | $1.5 \times 10^{-2}$ | 1 | 933 K | 300 K |

## 5. Influence Analysis of Key Parameters in Simulation

### 5.1. Verification of Blast Model

Figure 3 shows the empirical data and numerical data about the peak overpressure along with scaled distance of the explosion in air. By comparison, simulation data is almost the same as the empirical data [33–35], and the deviation is within the permissible scope of the project. Certain deviation between the simulation result and the empirical formula is mainly becausethe empirical formula is obtained by fitting according to the test results, while the simulation model is a relatively ideal result obtained by the calculation of the aerodynamic equation, and there are certain differences between the test conditions and the simulation models.

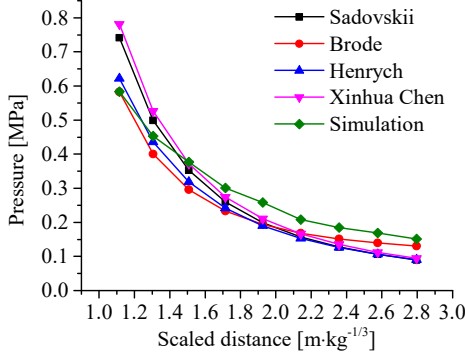

**Figure 3.** Comparison of theoretical and numerical data of the explosion in air.

### 5.2. Influence of Time Step on Simulation Results

In order to study the effect of time step on the peak overpressure, seven different time steps within the range from 0.1 μs to1.5 μs are simulated with the same mass of TNT explosive, namely $M_T$ = 51,151 kg, which is the equivalent TNT mass of the whole rocket. Four gauges with a vertical distance of 20 m to the explosive source are selected, whose parameters, including the coordinates and scaled distance, are shown in Table 6.

The peak overpressure of the blast wave on the crew module is affected by the time step in a way. As are shown in Figure 4, the peak overpressures decrease with the increase of scaled distance at different time steps. The curves of peak overpressure with scaled distance are not consistent with different time steps. When the time step grows to 1 μs, the overpressure curve deviates significantly from other curves, indicating that there is a large error in the simulation results. When the time step is less than 1 μs, several curves are close, indicating that the peak overpressure tends to be stable. Therefore, the time step of 1 μs is chosen in this paper because the calculation efficiency can be improved under the premise of accurately reflecting the calculation results.

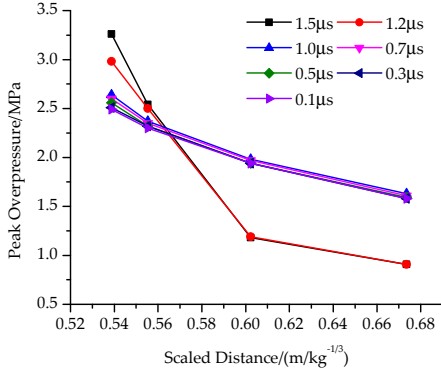

**Figure 4.** Peak overpressure versus scaled distance under different step size.

## *5.3. Influence of Grid Sizeon Simulation Results*

The accuracy of numerical results is highly sensitive to the grid size [36]. Therefore, it is necessary to analyze the convergence of grid size in numerical simulation to determine the appropriate grid size. In order to study the influence of grid size on numerical calculation results, seven kinds of grid sizes [37] are selected that range from 8 cm to 20 cm. In addition, the gradient grid is used to divide the air computing domain into seven parts, with grid sizes from 8 cm to 15 cm. A total of 3,241,252 grids are divided in the gradient grid model. Grid number is between the number of grid size of 8 cm and 10 cm, and the calculation time was also between them, as is shown in Table 10.

As can be seen from Figure 5, the effect of grid size on numerical calculation results depends on the scaled distance. When the scaled distance is less than $0.60 \text{ m·kg}^{-1/3}$, grid size has great influence on the calculation accuracy. For example, when scaled distance is $0.53878 \text{ m·kg}^{-1/3}$, peak overpressure of the model with grid size of 8 m is 5.22 MPa, while that of the model with grid size of 20 cm is 2.39 MPa. The former is 2.18 times the later. With the increase of scaled distance $R$, the influence of grid size on the results decreases gradually. What is more, for the same scaled distance, as the grid size decreases, the peak overpressure increases graduallyand finally tends to be stable. This is because the smaller the mesh, the better it responds to pressure changes. When the scaled distance is greater than $0.60 \text{ m·kg}^{-1/3}$, the peak overpressure of models with different grid sizes tends to be consistent.

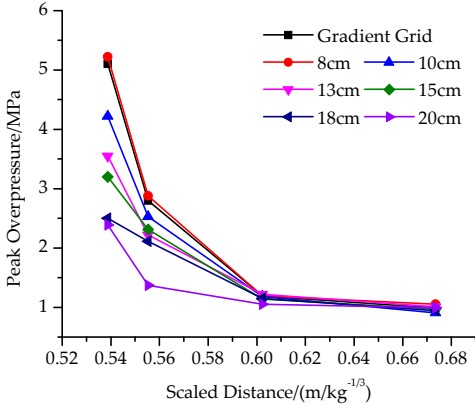

**Figure 5.** Peak overpressure versus scaled distance under different grid sizes.

**Table 10.** Grids number and calculation time of different grid sizes when time step is 1 μs.

| Grid Size/cm | Grids Number | Calculation Time/s |
|:---:|:---:|:---:|
| 20 | $3.40 \times 10^5$ | $3.30 \times 10^3$ |
| 18 | $4.19 \times 10^5$ | $3.98 \times 10^3$ |
| 15 | $8.04 \times 10^5$ | $8.01 \times 10^3$ |
| 13 | $1.19 \times 10^6$ | $1.78 \times 10^4$ |
| 10 | $1.30 \times 10^6$ | $2.52 \times 10^4$ |
| gradient grid | $3.24 \times 10^6$ | $3.06 \times 10^4$ |
| 8 | $4.90 \times 10^6$ | $3.76 \times 10^4$ |

Therefore, the grid size greatly influences the calculation results of the model. The gradient grid can improve the computational efficiency under the premise of accurately reflecting the propagation law of blast wave. The gradient grid will be used in the numerical simulation.

## 6. Results and Analysis

### *6.1. Analysis for Propagation Lawof Blast Wave*

The non-rocket model only considers the propagation law of the blast wave in the air, while the aluminum alloy structure in the rocket model will hinder the propagation of the blast wave and cause

complex reflection, flow around and superposition. According to the reflection theory of the blast wave, after the superposition of the incident wave and the reflected wave, the intensity of the blast wave will be greatly improved. In order to study propagation law of air blast wave of rocket explosion on launch pad, blast wave progress and its interaction with the rocket are obtained by simulating the model with and without a rocket. Take the stage-3 rocket explosion as an example to illustrate this problem.

Figure 6 shows the strain contours at different moments of stage-3 explosion, reappearing rocket damage process under the action of air blast wave after the explosion [38]. A strong chemical reaction takes place after detonation, resulting in a sharp expansion and diffusion in all directions of the high temperature and high-pressure explosive products, and the blast wave propagates outward in the form of a spherical wave. When $t = 0.6$ ms, the blast wave propagates to the rocketskin, causing the ellipsoid deformation of rocket structure. When $t = 1.2$ ms, the structure of the stage-3 is damaged, and the debris flow around in a spherical manner. With the propagation of blast wave, the structural damage of the body gradually extends to the upper and lower sides, and the area of structural damage gradually increases. When $t = 5.1$ ms, structure of almost the whole stage-3 and most of stage-2 are destroyed. When $t = 6.3$ ms, blast wave propagates to the crew module and causes certain damage to it. According to the theory of mechanical strength of materials, stress of aluminum alloycaused by blast wave exceeds the yield limit of the material, resulting in plastic deformation and failure [39].

Visualization of pressure contours available during the post-processing stage allows a better understanding of the complex process of blast pressure interaction with the rocket, as shown in Figure 7. It presents pressure contours after stage-3 explosion for a simulation time varying from 0.6 to 6.3 ms. Different contour images represent typical moments of blast wave propagation, and different pressure values are represented by different colours. At the blast onset, a rarefaction happens in the explosive charge volume, because of the rapid fluid expansion [40]. In the model without a rocket, the high-temperature and high-pressure explosive products produced by the explosive expand and spread rapidly around, and the blast wave spreads outward in the form of approximate spherical wave. With the passing of time, the wave front gradually broadens, but the spherical wave remains good. In the model with rocket, the propagation of blast wave becomes more complex. When $t = 1.2$ ms, the propagation of air blast wave is hindered by the rocket skin and the superposition of front and rear blast waves occurs, so the spherical blast wave begins to deform. When $t = 5.1$ ms, focusing phenomenon of blast wave appears in the rocket body, and the rest area still forms a stable blast wave to propagate outwards. The final waveform is conical in the upper and lower parts, and spherical in the middle. The higher overpressure points arein the tip of the cone-shaped blast wave. When $t = 6.3$ ms, the blast wave pressure concentrates at the crew module and has a destructive effect on it. The resulting pressure difference causes air flow and rarefaction wave.

It can be seen that the development process of the flow field is basically consistent with the actual physical process. The initial blast wave is not a spherical wave, but gradually approaches a spherical wave at a certain distance. At the moment of contact between the air blast wave and the rocket body, the airflow particles at the front of the blast wave are impeded, resulting in the reduction of velocity and the change of direction, which is superimposed with the later blast wave to form an enhanced blast wave inside the rocket body.

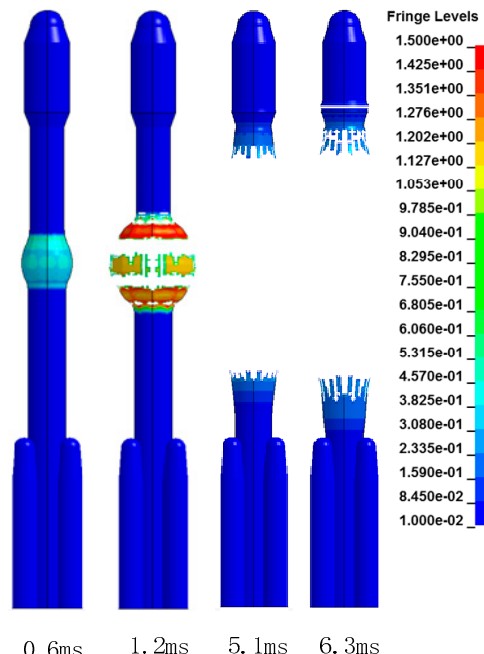

**Figure 6.** Strain contours at different moments of stage-3 explosion.

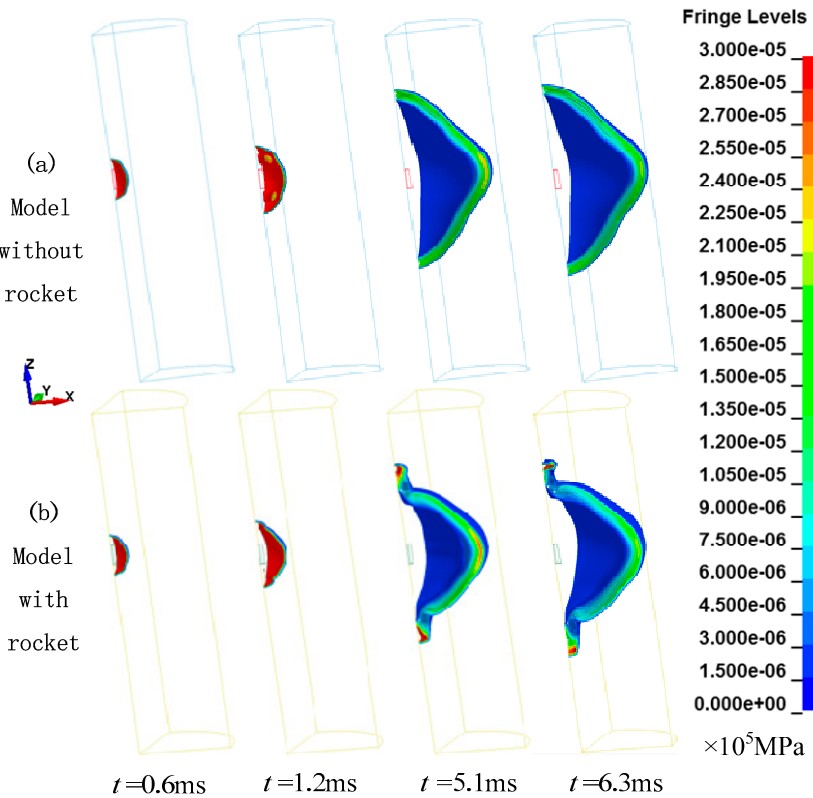

**Figure 7.** Pressure contours of air blast at different time after stage-3 explosion.

*6.2. Results of Different Explosion Scenarios*

Peak overpressuresof blast wave in different scenarios are shown in Table 11, which means damage to the crew module of different scenarios. As can be seen from Table 11, the most dangerous scenario for the crew module is Scenario 3, which represents full propellant involved in the explosion. The smaller the scaled distance is, the larger peak overpressure will be. Peak overpressure will be as

high as 35.84 MPa of Scenario 3, which will seriously damage the crew module. Similarly, the first explosion in Scenario 4 is the least harmful to the crew module. As its scaled distance is the biggest, its peak overpressure is the smallest, which is only 2.86 MPa.

All the simulation results exceed the low overpressures specified in the table, which will do great harm to the crew modules. Therefore, in order to prevent crew hazards, necessary protective measures should be taken for the crew modules.

**Table 11.** Peak overpressures of blast wave for different stages involved in explosion.

| Involved in Explosion | TNT Mass (kg) | Distance (m) | Scaled Distance (m·kg$^{-1/3}$) | Peak Overpressure (MPa) |
|---|---|---|---|---|
| whole rocket | $5.12 \times 10^4$ | 20 | $5.39 \times 10^{-1}$ | $3.58 \times 10^1$ |
| stage-1, 3 and boosters | $4.62 \times 10^4$ | 20 | $5.57 \times 10^{-1}$ | $3.19 \times 10^1$ |
| stage-3 | $1.11 \times 10^4$ | 15 | $6.73 \times 10^{-1}$ | $2.07 \times 10^1$ |
| stage-1, 2 and boosters | $4.01 \times 10^4$ | 36.5 | 1.07 | 4.97 |
| stage-2 | $4.98 \times 10^3$ | 20 | 1.17 | 3.99 |
| stage-2 and boosters | $2.14 \times 10^4$ | 37 | 1.33 | 3.14 |
| stage-1 | $1.87 \times 10^4$ | 36 | 1.35 | 2.86 |

*6.3. EnhancementEffects on Peak Overpressure of Model with aRocket*

The rocket model has a marked effect on the strength of the blast wave. For the whole rocket explosion scenario, the peak overpressure of model without a rocket is 2.09 MPa and that of model with a rocket is 35.84 MPa. The latter is 17 times of the former. Therefore, the influence of a rocket model on simulation results cannot be ignored.

Models with a rocket and without a rocket are established for each explosion scenario. The peak overpressure acting on crew module is taken respectively, and the influence of rocket body on the blast wave propagation is studied by comparing the results. It is assumed that in the model with and without a rocket, the peak overpressure of the blast wave acting on the crew module is $P_R$ and $P_A$ respectively, and the pressure enhancement factor $\alpha = P_R/P_A$ is defined. Under different scenarios, the peak overpressures and the pressure enhancement factors are shown in Table 12, and the pressure enhancement factor curve with scaled distance is shown in Figure 7.

As can be seen from Figure 8, $\alpha$ generally decreases with the scaled distance increases. When the scaled distance is less than 1.0 m/kg$^{-1/3}$, $\alpha$ decreases significantly with the increase of the scaled distance, from the highest 17.15 to 3.91. However, when the scaled distance is greater than 1.0 m/kg$^{-1/3}$, $\alpha$ does not change significantly with the scaled distance, and is stable at about 4.

The rocket model can enhance the peak overpressure of blast wave. The factor is about 4 times in far scaled distance, while it is greater than 4 times in near scaled distance and up to 17 times. Therefore, the rocket model cannot be ignored when studying the effect of blast wave on the crew module.

**Table 12.** Comparison of peak overpressures of the model with and without a rocket.

| Involved in Explosion | $P_R$ (MPa) | $P_X$ (MPa) | $\alpha$ |
|---|---|---|---|
| whole rocket | 35.84 | 2.09 | 17.15 |
| stage-1, 3 and boosters | 31.90 | 1.98 | 16.11 |
| stage-3 | 20.73 | 1.65 | 12.56 |
| stage-1, 2 and boosters | 4.97 | 1.27 | 3.91 |
| stage-2 | 3.99 | 1.01 | 3.95 |
| stage-2 and boosters | 3.14 | 0.80 | 3.92 |
| stage-1 | 2.86 | 0.71 | 4.03 |

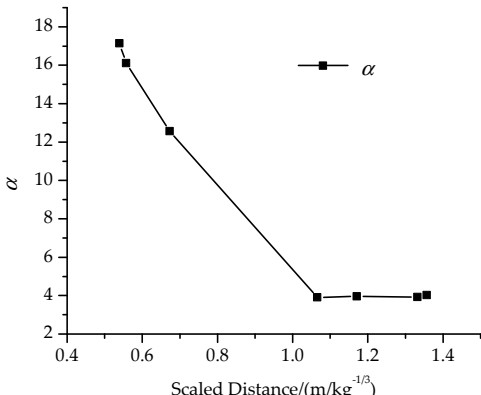

**Figure 8.** The pressure enhancement factor curve with scaled distance.

*6.4. PredictionFormula of Peak Overpressure*

The peak overpressure data with different scaled distances are fitted to obtain the prediction formula of peak overpressure applicable to the explosion mode on launch pad. The first-order attenuation exponential function in the form of Equation (9) is used for fitting, and the fitting Equation (9) and the fitting curve in Figure 9 are obtained.

Equation (9) describes the peak overpressure of blast wave acting on the crew modules as a function of the scaled distance and it can be used to predict the damage effect on crew modules of different stages explosions. As can be seen from Figure 9, when the scaled distance is small, the peak overpressure is very high. With the increase of the scaled distance, the peak overpressure decays rapidly and eventually tends to be flat. According to the calculated scaled distance, the peak overpressure of blast wave acting on the crew module can be obtained by consulting Figure 9, which greatly facilitates the prediction of the explosion hazard of the crew module.

$$\Delta P = 475.56 * \exp(-\bar{r}/0.20) + 2.26 \tag{9}$$

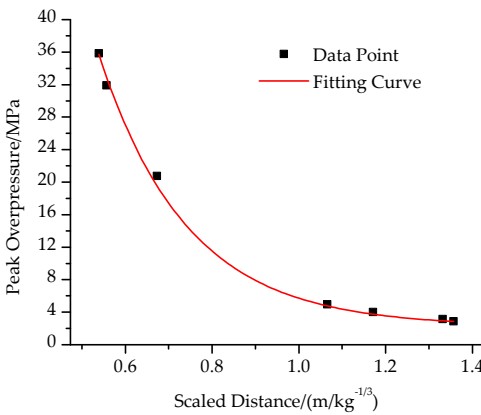

**Figure 9.** Fitting curve of the peak overpressures of blast wave versus scaled distances.

## 7. Conclusions

A numerical method for simulating the rocket explosion on the launch pad was developed in order to study the propagation law of blast wave and quantify the impact on the crew modules. After the analysis and discussion above, the following main conclusions were drawn:

(1) The final blast waveform of explosion model with rocket is conical in the upper and lower, and spherical in the middle.

(2)　In different scenarios of rocket explosion on launch pad, the third-stage explosion is the most harmful to the crew module, and the first-stage explosion is the least. In order to prevent crew hazards, necessary protective measures should be taken for the crew modules.

(3)　The rocket model has a marked effect on explosion strength. When the scaled distance is less than 1.0 m/kg$^{-1/3}$, the maximum of the pressure enhancement factor is about 17 times, and the minimum is about 4 times. When the scaled distance is greater than 1.0 m/kg$^{-1/3}$, the pressure enhancement factor is about 4 times. Therefore, the rocket model cannot be ignored when studying the effect of blast wave on the crew module.

(4)　Prediction formula of peak overpressureacting on the crew module in the explosion mode on the launch pad is established. It can quickly predict the peak overpressure of the blast wave and evaluate the damage to the crew module of the rocket explosion on the launch pad.

Predicted results are expected to be useful for hazards assessment associated with rocket explosion on the launch pad. Better understanding of explosion overpressure would improve the ability of crew modules protection.

**Author Contributions:** Methodology, Y.W., H.W.; software, Y.W., B.Z.; formal analysis, C.C.; investigation, Y.W.; resources, Y.W., H.W., C.C.; data curation, Y.W.; writing—original draft preparation, Y.W.; writing—review and editing, Y.W.; visualization, Y.W.

**Funding:** This research received no external funding.

**Acknowledgments:** The authors greatly appreciate the comments from the reviewers, whose comments helped to improve the quality of the paper.

**Conflicts of Interest:** The authors declare no conflict of interest.

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
