# Peer review of "Prediction Method of Blast Wave Impact on Crew Module for Liquid Rocket Explosion on Launch Pad"

_applsci, doi:10.3390/app9193976_

Round 1

Reviewer 1 Report

The paper concerns evaluation of blast loads in the course of accidental explosion of the rocket liquid propellant. An overview of the previous works represents interesting and valuable contribution. The consideration is based on the TNT equivalence method. This extremely simplified approach does not take into account real chemical transformations as well as dynamics of ignition, flame propagation and possible transition to detonation. However, the advantage of the paper is that the TNT equivalence method is implemented together with LS-DYNA package. As known, LS-DYNA calculates properly stress and crush phenomena that is important for developers.

            The authors focus their attention on the peak pressure of blast loading. However, in many real cases impulse of pressure wave also plays an important role. Moreover, normally, the TNT equivalence by impulse is different from that by pressure. The comments on this issue would be appropriate.

            It is no doubt that LS-DYNA correctly predicts blast parameters from TNT explosion in open space. In the presented paper application of LS-DYNA computations to a rather complicated rocket structure demonstrates possibility of prediction of parameters of blast loading under accidental conditions.

Some technical notes should be addressed before final submission.

Page 5, line 6 “different masses in the United States [2-34]” References [2-34] contains along with US papers from China, India, Russia… The mention of US should be excluded for the correctness or references should be corrected.

Page 5, line 13  What is “the specific coefficients”?

Page 5, line 14   What means “certain type rocket”?

Page 14, Table 12    Format must be changed at least to 3 decimal digits

Author Response

Dear professor:

I am the corresponding author of the manuscript “Prediction method of blast wave impact on crew module for liquid rocket explosion on launch pad”(ID applsci-582889). We must thank you for the critical comments and constructive recommendations. We feel lucky that our manuscript went to you as the valuable comments from you not only helped us with the improvement of our manuscript, but also raised some thoughtful suggestions.

Based on your comments and suggestions, we have made significant modification on the original manuscript. We have asked for native English speakers to revise the paper before it is submitted this time. We hope the new manuscript will meet your magazine’s standard.

Here are our one-by-one responses to your comments.

Point 1: Page 5, line 6 “different masses in the United States [2-34]” References [2-34] contains along with US papers from China, India, Russia… The mention of US should be excluded for the correctness or references should be corrected.

Response 1: I'm really sorry for this confusion. When writing the paper, we used Microsoft Office Word software, and the references here should be [2-4].We used the annotation method of inserting endnotes, in which 3 was hidden. But the format of 3 was changed in the version you saw, resulting in 3 being displayed and becoming [2-34]. We apologize for the inconvenience caused to you.

Point 2: Page 5, line 13  What is “the specific coefficients”?

Response 2: In the calculation of explosive yield, the chart-reading-method is an important method. It describes the functional relationship between the explosive power and the mass of the propellant involved. As shown in figure 3, different propellant combinations have different multiplication coefficients. The explosive yield is obtained through multiplying the explosive power by the multiplication coefficient corresponding to the propellant combination.

Point 3: Page 5, line 14   What means “certain type rocket”?

Response 3: The “certain type rocket” refers to a certain type of rocket. The research object in this paper does not refer to a particular type of rocket. The research method is applicable to any type of rocket. It has been modified in revised version.

Point 4: Page 14, Table 12    Format must be changed at least to 3 decimal digits

Response 4: All the data of overpressure in table 12 have been changed at least to 3 decimal digits in revised version.

Please see the revised version for detail. We hope the revised version can meet your concern.

Sincerely yours,

Yan Wang.

Reviewer 2 Report

The paper is fairly well put together and does seem to address an area lacking in the literature. The authors are commended for that. However, I have several comments and concerns, which I shall endeavor to address by section. The most important concerns are reiterated at the end of this review.

OVERALL COMMENTS

The authors are recommended to get an additional editing pass from a native English speaker, while the paper is mostly well-written, there are several sections which are ambiguous, confusing, or in need of reworking. I will therefore not comment on individual grammar errors and simply call out the most confusing sections. Please reformat your tables. It is extremely difficult to read them. Text should be top-aligned in its boxes, and except for occasionally in the header row, left-aligned as well. Tables should also not wrap pages. Significant digits are abused throughout. Please use scientific notation (A×10^B) and settle on a reasonable number of significant digits. The scenarios discussed in Tables 5, 6, 11. and 13 are ill-defined. Please add a section which discusses these scenarios, why they were chosen, and what they are meant to represent. Please explain how structure in between explosion centers was accounted for. Please also discuss how secondary explosions were handled (if stage 2 explodes, does that trigger explosions of stages 1 and 3?). Additionally, defining the scenarios in their own section will allow you to remove redundant columns in other tables (such as saying what stages are involved each time). Was any uncertainty analysis performed? If not, why not?

ABSTRACT

On line 9, it seems that the phrasing "once the rocket explodes" is unnecessarily defeatist. Thanks to advancements in modern technology, not every rocket explodes.

1. INTRODUCTION

In the first sentence, I am not sure that it is accurate that manned spaceflight is playing an "increasingly" important role in space exploration. Please reword. The final sentence of the first paragraph (page 1, lines 30-32) are confusing. How many modes are being referred to? On page 2, line 28, is it possible that the authors are referring to "parametric" analysis? If not, please clarify. On page 2, lines 44-45, how is it that accidents are "especially" multidisciplinary? On page 2, lines 49-51, please cite or expand on the assertion that the TNT model is a good method for predicting potential explosion risk. On page 2, lines 1-2, the authors state that the experimental conditions are strict and repeatability poor for propellant explosion tests. If experiments are fickle, how do you know that the simulations conducted in this study are accurate?

2. EXPLOSION MODE ANALYSIS

Table 1 is extremely difficult to read, as noted above. Why is "Soviet union" in italics but Brazil and USA not? Additionally, please capitalize the first letter of the first word in each cell. Finally, recommend "X deaths" rather than "X people are dead".

3. DEFINITE OF EXPLOSIVE YIELD

What does "Definite" mean? How is Y (or upsilon) in Eq. (1) determined? By experiment? By physics and chemistry? The following paragraph (the first of page 5) may hint at this, but it is difficult to tell. I do not understand the second paragraph on page 5 (lines 10-15). This entire section is extremely confusing and needs clarification and explanation. I do not understand what is being communicated by Tables 3 or 4. Please simplify and clarify. It appears they have something to do with explosive yield, but they are VERY confusing. On page 5, line 20-21, as well as Fig. 1, how were the size parameters of the rocket determined? Does it model a specific rocket model? Is it meant to represent a generic average size? What mission profile does it represent?
On page 5, line 25, please explain why distances were scaled against explosive mass. Table 5 is extremely confusing. Please clean up and add units where necessary. In particular, what is "Combined blast yield"?

4. FINITE ELEMENT MODEL

On line 14 of page 6, what is the purpose of mentioning the trial calculations? If they are important to the analysis, please discuss them here. If they are not, it is not necessary to mention them. On page 7, line 1, what does the Φ refer to? It is used twice. If it is diameter, please use the diameter symbol (Unicode U+2300). On page 7, lines 2-3, please clarify what is meant by "1/4 symmetrical geometric model". Is it a 1/4-scale model? Just 1/4 (one quadrant) of the model cross-section? If a quarter-scale model (geometrically similar), please discuss how dynamic and kinematic similarity were preserved in the analysis. On page 7, line 8, parentheses are required around the denominator of the expressions: h0/(2r0). The expressions of Eq. (4) are unnecessary and should be deleted.
The left side of Fig. 5(a) is illegible in color. (It's better, but still not good, printed in black-and-white.) What is the purpose of Fig. 5(b)? On page 8, line 15, the authors mention a relative volume, but not what it is relative to. Please give the reference volume by which V is normalized.

5. INFLUENCE ANALYSIS OF KEY PARAMETERS

On page 10, line 28, the authors state that overpressures stabilized as they approached the gradient and 8 cm grids. Were grids smaller than 8 cm tested to confirm this?

6. RESULTS AND ANALYSIS

I do not understand why the authors chose to report results without a rocket model. Surely it is self-evident that the presence of structure will significantly affect the observed results (overpressures et al.). Why is it necessary to devote an entire subsection (6.3), multiple figures (Fig. 7(a) and 8) and Table 13 to this self-evident result? Are the results in Sec. 6.2 and Table 11 done with or without the rocket model? If without, why? Table 12 is irrelevant and should be removed, as should the first sentence of the second paragraph of Sec. 6.2 (lines 17-18 of page 13). If it is necessary to provide some comparison, do so in the text and cite Ref. [41]. The discussion in Sec. 6.4 can be expanded. Such a correlation is likely useful, but its inclusion here seems an afterthought. Equations (9) is irrelevant and should be deleted. Eq. (10) should be reformatted and its significant figures refined.

CONCLUDING REMARKS

Many of the above remarks are minor and easily resolved. However, the authors are particularly requested to substantively address the concerns regarding

A better explanation of the scenarios Dynamic and kinematic similarity (if results from a scale model are presented) The results of Sec. 6.2/Fig. 11 being with or without rocket model

Additionally, the authors are requested to discuss accuracy and uncertainty. Are there experimental results that their work can be compared to?

Author Response

Dear professor:

I am the corresponding author of the manuscript “Prediction method of blast wave impact on crew module for liquid rocket explosion on launch pad”(ID applsci-582889). We must thank you for the critical comments and constructive recommendations. We feel lucky that our manuscript went to you as the valuable comments from you not only helped us with the improvement of our manuscript, but also raised some thoughtful suggestions.

Based on your comments and suggestions, we have made significant modification on the original manuscript. We have asked for native English speakers to revise the paper before it is submitted this time. We hope the new manuscript will meet your magazine’s standard.

Here are our one-by-one responses to your comments.

Point 1: OVERALL COMMENTS:The authors are recommended to get an additional editing pass from a native English speaker, while the paper is mostly well-written, there are several sections which are ambiguous, confusing, or in need of reworking. I will therefore not comment on individual grammar errors and simply call out the most confusing sections. Please reformat your tables. It is extremely difficult to read them. Text should be top-aligned in its boxes, and except for occasionally in the header row, left-aligned as well. Tables should also not wrap pages. Significant digits are abused throughout. Please use scientific notation (A×10^B) and settle on a reasonable number of significant digits. The scenarios discussed in Tables 5, 6, 11. and 13 are ill-defined. Please add a section which discusses these scenarios, why they were chosen, and what they are meant to represent. Please explain how structure in between explosion centers was accounted for. Please also discuss how secondary explosions were handled (if stage 2 explodes, does that trigger explosions of stages 1 and 3?). Additionally, defining the scenarios in their own section will allow you to remove redundant columns in other tables (such as saying what stages are involved each time). Was any uncertainty analysis performed? If not, why not?

Response 1: All the tables in the paper have been reformatted according your advice. Significant digits scientific notation are used properly throughout the paper. These four scenarios are discussed in detail in table 2. Combined explosive yields and combined explosive center coordinates are acquired according to the centroid formula. Each stage of the rocket is set with explosive center, and the combined explosive center is calculated according to the centroid formula. Figure 2 depicts four typical explosion scenarios. Take scenario 1 as an example, stage-3 explodes first, which results in the explosion of the stage-1, stage-2 and boosters one after another.

The rocket is a complex and large system. There are many reasons that lead to the explosion of the rocket. So there's a lot of uncertainty about how the rocket explodes. This paper chooses several typical extreme scenarios as representatives to carry out research to reflect the general law.

Point 2: ABSTRACT: On line 9, it seems that the phrasing "once the rocket explodes" is unnecessarily defeatist. Thanks to advancements in modern technology, not every rocket explodes.

Response 2: “Once” has been replaced by “in case”.

Point 3: INTRODUCTION: In the first sentence, I am not sure that it is accurate that manned spaceflight is playing an "increasingly" important role in space exploration. Please reword. The final sentence of the first paragraph (page 1, lines 30-32) are confusing. How many modes are being referred to? On page 2, line 28, is it possible that the authors are referring to "parametric" analysis? If not, please clarify. On page 2, lines 44-45, how is it that accidents are "especially" multidisciplinary? On page 2, lines 49-51, please cite or expand on the assertion that the TNT model is a good method for predicting potential explosion risk. On page 2, lines 1-2, the authors state that the experimental conditions are strict and repeatability poor for propellant explosion tests. If experiments are fickle, how do you know that the simulations conducted in this study are accurate?

Response 3: The "increasingly" has been deleted. The final sentence of the first paragraph has been rewritten. One advantage of the numerical simulation method is that it is convenient for parametric analysis. The accident explosion is a complex multidisciplinary problem that involves combustion, fluid dynamics, thermodynamics, structure/fluid interaction, and shock physics for high-speed flows. The uncertainty and complexity of the accident are more prominent. Reference [23] has been cited to explain on the assertion that the TNT model is a good method for predicting potential explosion risk. In order to explain that the operation of explosion test is very complicated and risky, small equivalent explosion test can be carried out to verify the simulation results. But an explosion on the scale of a rocket explosion would be even more difficult.

Point 4: EXPLOSION MODE ANALYSIS: Table 1 is extremely difficult to read, as noted above. Why is "Soviet union" in italics but Brazil and USA not? Additionally, please capitalize the first letter of the first word in each cell. Finally, recommend "X deaths" rather than "X people are dead".

Response 4: The font of the "Soviet union" in Table 1 has been modified in the revised version. "X people are dead" has been replaced by "X deaths" in Table 1.

Point 5: DEFINITE OF EXPLOSIVE YIELD: What does "Definite" mean? How is Y (or upsilon) in Eq. (1) determined? By experiment? By physics and chemistry? The following paragraph (the first of page 5) may hint at this, but it is difficult to tell. I do not understand the second paragraph on page 5 (lines 10-15). This entire section is extremely confusing and needs clarification and explanation. I do not understand what is being communicated by Tables 3 or 4. Please simplify and clarify. It appears they have something to do with explosive yield, but they are VERY confusing. On page 5, line 20-21, as well as Fig. 1, how were the size parameters of the rocket determined? Does it model a specific rocket model? Is it meant to represent a generic average size? What mission profile does it represent?

On page 5, line 25, please explain why distances were scaled against explosive mass. Table 5 is extremely confusing. Please clean up and add units where necessary. In particular, what is "Combined blast yield"?

Response 5: "Definite" has been replaced by "Definition". Y is determined by experiment carried out on different propellants of different masses in the United States. There are two methods, those are table-lookup-method and chart-reading-method. You can use table 3 for table-lookup-method, and table 4 for chart-reading-method. You can get the explosive yield from table 3 by using table-lookup-method. The explosive yield is obtained through multiplying the explosive power by the multiplication coefficient corresponding to the propellant combination. It model a specific rocket model. The rocket is settled on the launch pad before launching. As is shown in reference [23], there is a functional relationship between peak overpressure and scaled distance. You can get the peak overpressure from the scaled distance by using theoretical formulas. Combined explosive yields and combined explosive center coordinates are acquired by Eq. (2) and (3), on the basis of the explosion yields of each stage.

Point 6: FINITE ELEMENT MODEL: On line 14 of page 6, what is the purpose of mentioning the trial calculations? If they are important to the analysis, please discuss them here. If they are not, it is not necessary to mention them. On page 7, line 1, what does the Φ refer to? It is used twice. If it is diameter, please use the diameter symbol (Unicode U+2300). On page 7, lines 2-3, please clarify what is meant by "1/4 symmetrical geometric model". Is it a 1/4-scale model? Just 1/4 (one quadrant) of the model cross-section? If a quarter-scale model (geometrically similar), please discuss how dynamic and kinematic similarity were preserved in the analysis. On page 7, line 8, parentheses are required around the denominator of the expressions: h0/(2r0). The expressions of Eq. (4) are unnecessary and should be deleted.

The left side of Fig. 5(a) is illegible in color. (It's better, but still not good, printed in black-and-white.) What is the purpose of Fig. 5(b)? On page 8, line 15, the authors mention a relative volume, but not what it is relative to. Please give the reference volume by which V is normalized.

Response 6: “After several trial calculations attempt” has been replaced by “In order”. Φ refers to the diameter and the diameter symbol (Unicode U+2300) has been used in the paper. "1/4 symmetrical geometric model" means one quadrant of the model cross-section. Pay attention to the symmetry of the model and adopt the simplification on the model. In LS-DYNA software, when the shape, material and load of the physical system are symmetrical, it is possible to model and analyze only the representative parts or sections of the actual structure, and then map the results to the entire model, so as to obtain the results with the same accuracy and reduce the time spent in modeling and calculation.

Point 7: INFLUENCE ANALYSIS OF KEY PARAMETERS: On page 10, line 28, the authors state that overpressures stabilized as they approached the gradient and 8 cm grids. Were grids smaller than 8 cm tested to confirm this?

Response 7: Grids smaller than 8 cm were tested to confirm this.

Point 8: RESULTS AND ANALYSIS: I do not understand why the authors chose to report results without a rocket model. Surely it is self-evident that the presence of structure will significantly affect the observed results (overpressures et al.). Why is it necessary to devote an entire subsection (6.3), multiple figures (Fig. 7(a) and 8) and Table 13 to this self-evident result? Are the results in Sec. 6.2 and Table 11 done with or without the rocket model? If without, why? Table 12 is irrelevant and should be removed, as should the first sentence of the second paragraph of Sec. 6.2 (lines 17-18 of page 13). If it is necessary to provide some comparison, do so in the text and cite Ref. [41]. The discussion in Sec. 6.4 can be expanded. Such a correlation is likely useful, but its inclusion here seems an afterthought. Equation (9) is irrelevant and should be deleted. Eq. (10) should be reformatted and its significant figures refined.

Response 8: It can be seen from the result analysis, the rocket model have a markedly effect on the strength of blast wave. For the whole rocket explosion scenario, the peak overpressure of model without rocket is 2.09MPa and that of model with rocket is 35.84MPa. The later is 17 times of the former. Therefore, the influence of rocket model on simulation results cannot be ignored. The results in Sec. 6.2 and Table 11 are done with the rocket model. Table 12 and the first sentence of the second paragraph of Sec. 6.2 have been deleted from the paper. Eq. (10) describes the peak overpressure of blast wave acting on the crew modules as a function of the scaled distance and it can be used to predict the damage effect on crew modules of different stages explosions. The discussion in Sec. 6.4 has been expanded. Equation (9) has been deleted. Eq. (10) has been reformatted and its significant figures has been refined.

Point 9: CONCLUDING REMARKS: A better explanation of the scenarios Dynamic and kinematic similarity (if results from a scale model are presented) The results of Sec. 6.2/Fig. 11 being with or without rocket model.

Response 9: "1/4 symmetrical geometric model" means one quadrant of the model cross-section, not a 1/4-scale model. Pay attention to the symmetry of the model and adopt the simplification on the model. In LS-DYNA software, when the shape, material and load of the physical system are symmetrical, it is possible to model and analyze only the representative parts or sections of the actual structure, and then map the results to the entire model, so as to obtain the results with the same accuracy and reduce the time spent in modeling and calculation. The results in Sec. 6.2 and Table 11 are done with the rocket model.

Point 10: Additionally, the authors are requested to discuss accuracy and uncertainty. Are there experimental results that their work can be compared to?

Response 10: The rocket is a complex and large system. There are many reasons that lead to the explosion of the rocket. So there's a lot of uncertainty about how the rocket explodes. This paper chooses several typical extreme scenarios as representatives to carry out research to reflect the general law.

The accident explosion is a complex multidisciplinary problem that involves combustion, fluid dynamics, thermodynamics, structure/fluid interaction, and shock physics for high-speed flows. The uncertainty and complexity of the accident are more prominent. Reference [23] has been cited to explain on the assertion that the TNT model is a good method for predicting potential explosion risk. In order to explain that the operation of explosion test is very complicated and risky, small equivalent explosion test can be carried out to verify the simulation results. But an explosion on the scale of a rocket explosion would be even more difficult.

Please see the revised version for detail. We hope the revised version can meet your concern.

Sincerely yours,

Yan Wang.

Reviewer 3 Report

The paper presents and discusses results from numerical simulations of blast wave propagation in rocket-like geometrical configurations. The numerical results are used to deduce a simple empirical equation for peak over-pressures as a function of the reduced distance to a TNT-equivalent explosion source.
The results are relevant to the field of launcher design and safety analysis.
The well-established commercial LS-DYNA software was used for the simulations. Adequate assessment of the influence of numerical discretization parameters (grid density, time step) is shown.
The common concept of a TNT-equivalent explosion source was chosen.

The numerical analyses and simulations of the blast wave propagation were based on a very much simplified model of the structure. This represents the biggest weakness of the presented work. The reduction of the finite-element-model to a cylindrical domain without any lids, inter-stage structures, bulkheads, tanks, heavy structures like engine-assemblies enables free propagation of the blast wave inside the structure in axial direction. It is not clear, whether the large over-pressures due to the focusing of the "internal" blast wave would occur also in a real configuration. The significance of the paper would be substantially stronger if the influence of axial obstacles would have been included.

Further, it is mentioned a couple of times that the simulations are performed for the rocket on the launch pad. The presence of the launch pad can nowhere be seen in the results (e.g. Fig 7). A clarification of how the launch pad affects the results is missing.

Suggestions for minor revisions:

Page 3, line 7: First appearance of "LS-DYNA". Would be nice to have a reference or to mention that this is a commercial FE-package.
Page 5, Table 3: It is not clear where the numbers come from. Ref [22] only or Refs [2-34]? Suggestion: If the numbers in Table 5 are from different sources, mention the respective reference in the Table next to the numbers.
Page 5, Table 3: What is the "Specific coefficient" ? Is it needed for the further analyses ?
Page 6, Table 5: "Blast yield": this is called "explosive yield" in the rest of the paper, maybe use consistent notations.
Page 7, line 1: It appears the the greek letter Phi should specify a diameter (or radius). THis should be replaced by a better notation. Why is lenght of the crew module only 0.1 m?
Page 13, Fig. 7: Units are missing in the legend.

Generally: The English needs revision, especially in section 5.3

Author Response

Dear professor:

I am the corresponding author of the manuscript “Prediction method of blast wave impact on crew module for liquid rocket explosion on launch pad”(ID applsci-582889). We must thank you for the critical comments and constructive recommendations. We feel lucky that our manuscript went to you as the valuable comments from you not only helped us with the improvement of our manuscript, but also raised some thoughtful suggestions.

Based on your comments and suggestions, we have made significant modification on the original manuscript. We have asked for native English speakers to revise the paper before it is submitted this time. We hope the new manuscript will meet your magazine’s standard.

Here are our one-by-one responses to your comments.

Point 1: Page 3, line 7: First appearance of "LS-DYNA". Would be nice to have a reference or to mention that this is a commercial FE-package.

Response 1: Reference 20 has been added in line 102 in the revised version.

Point 2: Page 5, Table 3: It is not clear where the numbers come from. Ref [22] only or Refs [2-34]? Suggestion: If the numbers in Table 5 are from different sources, mention the respective reference in the Table next to the numbers.

Response 2: Sorry for this confusion. We made an error in the citation, which should have referred to the data in reference 23 for table 3. It has been changed in the revised version.

Point 3: Page 5, Table 3: What is the "Specific coefficient" ? Is it needed for the further analyses ?

Response 3: In the calculation of explosive yield, the chart-reading-method is an important method. It describes the functional relationship between the explosive power and the mass of the propellant involved. As shown in figure 3, different propellant combinations have different multiplication coefficients. The explosive yield is obtained through multiplying the explosive power by the multiplication coefficient corresponding to the propellant combination.

Point 4: Page 6, Table 5: "Blast yield": this is called "explosive yield" in the rest of the paper, maybe use consistent notations.

Response 4: Sorry for this confusion. This is our mistake in writing the paper. All the "Blast yield" of the whole passage have been replaced by "explosive yield" in the revised version.

Point 5: Page 7, line 1: It appears the the greek letter Phi should specify a diameter (or radius). THis should be replaced by a better notation. Why is length of the crew module only 0.1 m?

Response 5: h0/2r0 has been replaced by the greek letter φ in the revised version. The size of the crew module is unknown. The crew module established here is mainly to obtain the shock wave parameters acting on it, and the size does not affect the results.

Point 6: Page 13, Fig. 7: Units are missing in the legend.

Response 6: The unit of the legend is ×105MPa, and it has been added in Fig. 7 in the revised version.

Point 7: Generally: The English needs revision, especially in section 5.3

Response 7: The English of the whole passage has been revised, especially in section 5.3.

Please see the revised version for detail. We hope the revised version can meet your concern.

Sincerely yours,

Yan Wang.
